# The Occurrence and Meta-Analysis of Investigations on Intestinal Parasitic Infections Among Captive Wild Mammals in Mainland China

**DOI:** 10.3390/vetsci12020182

**Published:** 2025-02-18

**Authors:** Xueping Zhang, Hongyu Zhou, Lina Ye, Jiayu Shi, Huiling Zhang, Tangjie Zhang

**Affiliations:** 1Institute of Comparative Medicine, College of Veterinary Medicine, Yangzhou University, Yangzhou 225009, China; zhangxp1995@foxmail.com (X.Z.); zhou2096177403@foxmail.com (H.Z.); 201901123@stu.yzu.edu.cn (L.Y.); 18251238559@163.com (J.S.); 2Jiangsu Co-Innovation Center for Prevention and Control of Important Animal Infectious Diseases and Zoonoses, Yangzhou 225009, China; 3Independent Researcher, New York, NY 11355, USA

**Keywords:** meta-analysis, captive wild mammals, intestinal parasite infections, mainland China, cross-sectional studies

## Abstract

To ensure the welfare of captive wild animals and public health, it is essential to assess and monitor the gastrointestinal parasitic infection status of captive wild mammals. This study presents the first meta-analysis and systematic review to evaluate the prevalence of gastrointestinal parasitic infections among captive wild mammals in mainland China and the associated risk factors. A total of 29 eligible studies were included in this comprehensive analysis. The findings revealed a high prevalence of gastrointestinal parasitic infections among captive wild mammals in mainland China, with nematodes identified as the most dominant parasite group. Among taxonomic orders, primates exhibited the highest infection rates, followed by Artiodactyla and Rodentia, whereas Proboscidea showed the lowest prevalence. Seasonally, infection rates peaked during summer and winter, likely due to favorable environmental conditions and behavioral patterns of captive animals. Furthermore, infection prevalence was lower in economically developed regions, suggesting that improved sanitation and management practices play a critical role in mitigating parasite transmission. We emphasize the need for a sanitation management program and regular antiparasitic treatment in captive wild animals.

## 1. Introduction

China is home to approximately one of tenth of the world’s total wild mammal species [1]. However, factors such as environmental changes, poaching, and illegal trade pose significant threats to wildlife, increasing the risk of animal-borne diseases [2,3]. Since the rise of human civilization, 83% of wild mammal species have become extinct. From mice to elephants, only one-sixth of wild mammal species remain [4]. To protect wildlife and maintain biodiversity, some wild mammals are kept within human-controlled environments. Captive wildlife mainly includes animals from zoos, conservation centers, artificial breeding facilities, parks, etc. [5].

Parasites play a critical role in shaping ecological communities. They are important in food webs, competitive interactions, biodiversity patterns, and the regulation of keystone species [6,7]. The gut microbiota of wild mammals, which includes the normal intestinal flora and microorganisms encountered in the environment, is closely associated with the physiology and health of individual animals. Healthy ecosystems often have a high diversity of parasites. Parasitic infections are ubiquitous in wildlife, livestock, and human populations [8]. However, the limited enclosure sizes in captive environments lead to smaller activity spaces compared to the wild, resulting in higher stocking densities. Consequently, they are more susceptible to disease invasion and transmission than their counterparts in the wild. Research indicates that parasitic diseases are among the primary afflictions endangering wild mammals [9,10]. These diseases can lead to malnutrition, tissue and organ damage, and toxic effects. The rise in parasitic diseases in captive wild mammals may also impact on human health and well-being [11,12]. Captive wild mammals are also a source of parasitic diseases for humans and livestock. Zoonotic parasites are likely to be transmitted from animals to human due to direct contact between animal caretakers or tourists and wild animals.

Research on parasitic diseases in wildlife remains underdeveloped in mainland China, with only a few systematic studies conducted [13,14]. This study aimed to assess the prevalence and associated risk factors of intestinal parasites in captive wild mammals in mainland China through a systematic review and meta-analysis.

## 2. Materials and Methods

### 2.1. Search Strategy

This study followed the PRISMA guidelines (Preferred Reporting Items for Systematic Reviews and Meta-Analyses) [15]. The PRISMA checklist was used to ensure that all relevant information was included in the analysis (see Appendix A).

Literature searches were conducted using four databases: China National Knowledge Infrastructure (CNKI), Wanfang Data Knowledge Service Platform, PubMed, and Web of Science. The Chinese search terms were “parasites” or “intestinal parasites”, “captive wild animals”, “Zoo” or “Park”. The English search terms were “parasites” or “intestinal parasites”, “captive wild animal”, “Zoo” or “Park” and “China”. The search covered the period from the inception of the databases to December 2024. Additionally, the reference lists of each retrieved article were manually searched to identify further relevant articles.

### 2.2. Inclusion and Exclusion Criteria

After removing duplicate and review articles, preliminary screening was conducted based on the title and abstract of each article to further exclude irrelevant studies.

Full texts were obtained and reviewed to determine if they met the following criteria: ① The target subjects were captive wild mammals. ② The detected parasites were intestinal parasites. ③ Clear information on sample collection time and source. ④ Explicit detection methods (molecular or microscopic techniques). ⑤ Cross-sectional studies.

If a disagreement or uncertainty arose about the eligibility of a study, the discrepancies were resolved by a third party or through discussion.

### 2.3. Data Extraction

Two reviewers independently extracted and recorded specific information from the selected literature. A data collection form was created using Microsoft Excel (2016 for Windows), including: ① Study characteristics: literature title, first author, publication date. ② Study methods: sampling time, sample type, detection methods. ③ Animal classification (host): order. ④ Name and classification of detected parasites (protozoan or nematode or trematode or cestoda). ⑤ Sample size and positive size.

### 2.4. Literature Quality Assessment

Cross-sectional studies in animals differ from randomized clinical trials; their systematic evaluation methods are not as mature, and they lack control groups. Therefore, the systematic evaluation methods for animal cross-sectional clinical trials were adjusted based on Cochrane quality evaluation criteria, and incorporated the high-quality project assessment scale developed by Munn et al. [16] and the modified quality assessment checklist by Ding et al. [17], we adjusted the systematic evaluation method for this study. The assessment items for evaluating the risk of bias in the included studies are as follows: ① Is the research question/objective clearly described and stated? ② Is the source of wild mammals clearly stated? ③ Is the sampling time clearly stated? ④ Is the parasitic detection method clearly and described in detail? ⑤ Is there detailed information on the detected intestinal parasites, including specific parasite names and classifications? ⑥ Is the classification of animals clearly specified?

### 2.5. Statistical Analysis

Data collection and organization were conducted using Microsoft Excel (2016 for Windows). Data processing and statistical analysis in this meta-analysis were performed using Stata 15.1 software. Given the presence of extreme values in the included single-group rate data (100% or 0%), forest plots were analyzed using the metaprop package [18], with data transformed using the Freeman-Tukey double arcsine transformation [19,20]. The formula is as follows: t = arcsin[sqrt {r/(n + 1)}] + arcsin[sqrt {(r + 1)/(n + 1)}], where (t) represents the transformed infection rate, (r) represents the number of positive cases, and (n) represents the sample size. The standard error (se) of (t) is calculated as se(t) = sqrt{1/(n + 0.5)} Proportional reverse transformation was performed using the following formula: p = (sin(t/2))^2^.

Heterogeneity assessment of infection rates in the included studies was conducted using I^2^ and Q tests. If (*p* ≥ 0.10) and (I^2^ < 50%), it indicates low or no statistical heterogeneity among the effect sizes of the studies, and a fixed-effects model should be used for analysis [21]. If (*p* < 0.10) and (I^2^ ≥ 50%), it indicates significant statistical heterogeneity among the study effects, and a random-effects model, specifically the Der-simonian-Laird (D-L) method, should be used to synthesize all estimated values of the combined infection rate and its 95% confidence interval.

Through grouping based on potentially relevant features, further investigation of heterogeneous potential sources can be conducted. In this study, subgroup analysis was performed based on season, year, parasite category, animal order and feeding habits.

Bias and sensitivity analysis were conducted in this study using funnel plots. In asymmetric funnel plots, the trim-and-fill method was used to interpolate potentially missing studies and estimate the corrected prevalence [22,23]. Furthermore, sensitivity analysis was conducted to test the robustness of the results obtained in this study [24], and heterogeneity factors were identified using subgroup analysis.

## 3. Results

### 3.1. Literature Retrieval and Basic Characteristics of Included Studies

Following the previously described search strategy, 314 articles published from 1 January 2000, to 1 December 2024, were retrieved from four databases. After excluding 45 duplicated articles from the databases and reviewing the remaining ones based on the inclusion criteria following the Cochrane handbook’s literature selection process, a total of 29 eligible studies were considered for meta-analysis. The selection process and results are illustrated in Figure 1.

These studies, published between 1 January 2007, and 1 December 2024, were analyzed. A total of 8421 captive wild mammals were included, ranging from 12 to 3349 individuals. Characteristics of the included studies are summarized in Table 1 and Appendix A.

### 3.2. Outcomes of Literature Quality Evaluation

Results of study quality are presented in Table 2. Based on six quality assessment criteria, the maximum score was 12 points. The mean ± standard deviation of the overall quality assessment scores was 10.63 ± 1.33. The median score was 11 with the score ranging from 6 to 12 points (Appendix A).

### 3.3. Prevalence Heterogeneity Analysis

Figure 2 presents the forest plot depicting the prevalence of intestinal parasite infection in captive wild mammals in China from 2007 to 2024.

The overall prevalence of intestinal parasitism in captive wild mammals in mainland China ranged from 9.8% to 100%, as reported in the included studies. Significant heterogeneity was observed among the studies (χ^2^ = 1045.578; *p* < 0.001; I^2^ = 97.322%). A random-effects model was applied to calculate the pooled prevalence, yielding an estimate of 53.9% (95% CI, 46.6–61.2%).

### 3.4. Subgroup Analysis

The subgroup analysis based on season, year, parasite category, feeding habits, order, and detection method is shown in Table 3. There was a significant difference in parasitic infection rates across seasons (*p* = 0.038). The parasitic infection rates ranked from highest to lowest were: summer 61.8% (95% CI, 51.2–71.9%),winter 61.6% (95% CI, 46–76.1%), autumn 50% (95% CI, 6.6–93.4%), spring 43% (95% CI, 24.4–62.6%), and year-round 42.6% (95% CI, 32–53.5%).

There is significant heterogeneity in infection rates across regions (*p* = 0.009). The highest infection rate was observed in the Northeast region at 71.2% (95% CI, 61.6–79.9%), followed by the Western region at 56.9% (95% CI, 47.4–87%), the Central region at 51.5% (95% CI, 38.7–64.2%), and the lowest at 45.9% (95% CI, 47.4–87%) in the Eastern region.

The infection rates of different species of parasites showed extremely significant differences (*p* = 0.0001). The highest infection rate was occurred in nematodes at 45.1% (95% CI, 37.2–53.1%), followed by protozoa at 27.4% (95% CI, 18.2–37.8%), trematodes at 5.5% (95% CI, 1.1–12.4%), and the lowest infection rate occurred in cestodes at 2.7% (95% CI, 1.3–4.6%).

A subgroup analysis was performed on animals from various orders. The analysis revealed significant differences in intestinal parasite infection rates among them (*p* = 0.004). Within the class Mammalia, the highest infection rate was observed in Primates at 66.5% (95% CI, 53.3–78.7%), followed by Artiodactyla at 59% (95% CI, 45.6–71.7%), Rodentia at 57.1% (95% CI, 8.8–98.6%), and the lowest occurred in the Proboscidea at 19.9%(95% CI, 3.9–41.4%) (Figure 3). The distribution of parasitic infections among different animal orders is illustrated in Figure 4. Overall, nematodes and protozoa exhibited the highest infection rates. Nematode infections were predominant in Primates, Rodentia, Artiodactyla, Perissodactyla, and Carnivora, whereas protozoan infections were more prevalent in Diprotodontia and Proboscidea. In contrast, trematodes and cestodes infection rates were comparatively low.

The subgroup analysis of detection methods revealed a significant difference between conventional methods and PCR (*p* = 0.012). The infection rate detected using conventional methods was 55.3% (95% CI, 47.1–63.4%), whereas PCR detected an infection rate of 42.9% (95% CI, 37.7–48.1%).

The feeding habits subgroup analysis showed no significant difference in infection rates among carnivores, herbivores, and omnivores (*p* = 0.971). The infection rates were 55.0% (95% CI, 43.6–66.1%) for carnivores, 56.6% (95% CI, 45.1–66.4%) for herbivores, and 55.5% (95% CI, 44.4–66.4%) for omnivores.

A review of sampling times indicated 11 articles from 2007 to 2012, seven articles from 2013 to 2018, and six articles from 2019 to 2024. Subgroup analysis by sampling time indicated no significant difference (*p* = 0.459).

### 3.5. Publication Bias and Sensitivity Analysis

A portion of the studies (15) fell outside the confidence interval as shown in the funnel plot (Figure 5), indicating publication bias among the included studies.

Given that publication bias can cause funnel plot asymmetry, an iterative method was used to estimate the number of missing studies. A new meta-analysis was then conducted using the trim-and-fill method (the Metatrim command from Stata 15.1 software) to determine the impact of publication bias on the research results [22,23]. After incorporating seven additional virtual studies, the meta-analysis was performed again, resulting in an adjusted *p*-value of 0.000 (Figure 6). Heterogeneity remained significant, suggesting that the adjusted results were statistically robust.

The results of the sensitivity analysis indicated that the prevalence of intestinal parasite infection remained within the 95% confidence interval (Figure 7). This suggests that the results of the meta-analysis were robust and not significantly affected by variations in study inclusion.

## 4. Discussion

Globally, 60% of terrestrial mammals are domesticated livestock, predominantly consisting of cattle and pigs. Humans account for 36% of the mammalian population, while wild mammals represent a mere 4%. Humans are highly efficient at exploiting natural resources, and across nearly all continents, hunting has driven many wild mammals to extinction, often for food or recreation [4]. To our knowledge, this is the first meta-analysis and systematic review on the prevalence of intestinal parasite infections in captive wild mammals in China. The combined infection rate in this study was derived from a comprehensive analysis of scientific publications on the prevalence of intestinal parasite infections in captive wild mammals in mainland China from 2007 to 2024. Our study estimated the intestinal parasite infection status of 8421 captive wild animal samples from 29 studies. The combined prevalence of intestinal parasite infection was 53.9% (95% CI, 46.6–61.2%). This was lower than the prevalence reported in the Rio de Janeiro Zoo (68.3%) in southeastern Brazil, Rabat Zoo (70%) in Morocco, and in Bangladesh (65.3%, 95% CI: 53.14–76.12%) [54,55,56]. However, it was higher than the prevalence reported in zoos in Slovenia (45%) [57]. This suggests that most captive wild mammals worldwide have a high prevalence of gastrointestinal parasitic infections.

Our study found that the infection rate of nematodes in captive mammals in mainland China was the highest, at 45.1% (95% CI, 37.2–53.1%). This high rate may be related to the variety and abundance of nematodes reported in studies conducted in mainland China, warranting further investigation. Mir et al. and Ferdous et al. also reported that in wild mammals, the prevalence of nematode infections was higher than that of other parasites [55,58]. Nematodes can be transmitted through soil, water sources, food, and direct contact, facilitating their spread in captive environments. Among various nematode species, roundworms, whipworms, and hookworms are zoonotic parasites [59,60,61], posing significant public health risks. Zoos should prioritize enhanced nematode control through effective deworming management.

Seasonality is one of the important factors affecting parasite infection of. Conditions such as warmth and humidity favor the development of eggs and larvae in the external environment, making infections more common in spring and summer [62,63]. The transmission of some parasitic diseases depends on intermediate hosts and vector arthropods. As a result, the transmission and infection seasons coincide align with the emergence of their vector arthropods, influencing the infection rate [64]. Morgan et al. demonstrated that climate strongly influences the free-living stages of gastrointestinal nematodes in sheep [65]. Both extreme heat and cold are detrimental to their development and survival. Chen’s research found that the infection rate of gastrointestinal parasites in Ili horses in Xinjiang is typically higher in spring than in autumn [66]. Similarly, Hu et al. found in wild giant pandas that the prevalence of parasite infections correlates with rising temperatures [67].

Our research findings show significant variations in gastrointestinal parasite infection rates across different seasons and regions within the same season. The highest infection rate was observed in summer at 61.8%, while winter also showed a relatively high infection rate of 61.6%. The high temperatures and humid conditions in summer create an optimal environment for the proliferation of mosquitoes, ticks, and other vectors, facilitating the expansion of intermediate hosts and transmission pathways, ultimately contributing to an increased parasite infection rate [68].

In captive environments, animals often exhibit reduced outdoor activities during winter, leading to more congregations within enclosures, reduced indoor ventilation compared to spring and autumn, and hygiene challenges. These factors collectively contribute to higher parasite infection rates during winter [69], a phenomenon potentially common among captive wild mammals in mainland China. Some parasites may also be more easily transmitted in winter [61]. Carlsson et al. found in experiments that, contrary to most parasitic nematodes, Marshallagia marshalli of Svalbard reindeer is transmitted during the Arctic winter [70]. Further research is needed on the impact of temperature on the infection rate of parasites in captive wild mammals.

China’s vast landmass exhibits significant geographical and climatic variations. Generally, the warm and humid climate of southern regions provides favorable conditions for parasite reproduction and transmission, leading to relatively high infection rates. In contrast, northern China is characterized by a temperate monsoon climate, while the northwest experiences an arid climate with scarce precipitation, both of which contribute to relatively lower parasite infection rates. However, the level of economic development in mainland China may exert a more significant influence on parasite infection [71]. Economically developed regions typically possess well-established infrastructure, including access to clean drinking water and improved sanitation. These regions also have better medical resources, such as specialized personnel, equipment, and medications, along with higher public awareness of health and hygiene [72]. Health education programs are more effectively implemented, and substantial investments are made in research related to parasite prevention and control [73]. In this study, the eastern regions of China, which are economically developed, exhibited the lowest parasite infection rate (45.9%). Conversely, the highest infection rate (71.2%) was observed in northeastern China, which may be attributed to lower levels of economic development. However, the relatively small sample size in this study necessitates further research to validate these findings.

This study found significant differences in infection rates among animals from different taxonomic orders. Specifically, the highest infection rate of gastrointestinal parasites occurred in the order Primates, at 66.5%. This rate is significantly higher than the rates reported for non-human primates in zoological institutions in France [74] (53.9%) and for primates at Zoo Negara in Malaysia [75] (54.5%). As social animals, primates frequently interact within and between groups, facilitating parasite transmission. In mainland China, increased interactions between zoo visitors and primates may further contribute to the spread of gastrointestinal parasites.

Additionally, the study found that the infection rate in Artiodactyla (59%) was higher than that in Perissodactyla (32.1%), consistent with findings from the Rio de Janeiro Zoo [54]. Variation in parasitic infection rates across different taxonomic orders may be influenced by differences in digestive physiology, social behaviors, environmental adaptability, and immune responses. For instance, Artiodactyla generally exhibit more complex social structures than Perissodactyla, which tend to have smaller group sizes and simpler social dynamics. Furthermore, Artiodactyla engage in rumination, and their multi-chambered stomachs harbor diverse microbial populations, potentially affecting parasite susceptibility [76]. Therefore, zoos should implement targeted parasite control measures tailored to the specific biological and behavioral characteristics of each animal order. With regard to dietary habits, several studies have demonstrated that herbivorous animals tend to have a higher prevalence of gastrointestinal parasite infections than carnivorous animals [55,77]. However, in our study, no significant difference in infection rates was observed among animals with different feeding habits, with only a marginal difference. This suggests that feeding habits of captive animals in mainland China do not significantly influence parasite infection rates.

The differences observed in gastrointestinal parasite infections among different taxa are likely due to various factors, such as regional differences in husbandry practices and habits, sanitation management programs, and individual variations among animals. It is recommended that zoos implement scientific hygiene management plans, including regular monitoring, and treat gastrointestinal parasites in captive wild mammals.

There are significant differences between traditional detection methods and molecular detection in terms of detection results. Historically, the detection of intestinal parasites in literature mainly relied on microscopic examination of fecal samples. Microscopic parasite examination is highly dependent on the operator’s experience and technical skills. Such reliance can lead to false positives due to subjective judgment [78,79]. In the literature we reviewed, PCR has been used to detect parasites since its introduction in 2021. New diagnostic techniques offer more accurate, sensitive, and specific results. As a result, traditional detection methods often identify higher infection levels than PCR methods, which were introduced later. Considering the advancements in molecular diagnostic technology, we recommend prioritizing molecular detection methods or experienced clinical personnel for the diagnosis of gastrointestinal parasites in captive wild mammals, provided that the necessary resources and conditions are available.

This study has several limitations: 1. The current methodology does not comprehensively capture all gastrointestinal parasitic infections in captive wild mammals in mainland China. 2. The literature included in the study lacked comprehensive information on intestinal parasitic infections. This limitation suggests that there may be other factors influencing the results may have been overlooked. 3. Age was not investigated for its impact on gastrointestinal parasite infections in wild mammals. This omission could overlook significant nuances in the dynamics of parasite infections across different age groups. 4. This study covers 15 provincial-level administrative regions in China. Although it represents a wide range of regions in mainland China, its geographical coverage remains incomplete, with less representation from the northeast region in the literature.

## 5. Conclusions

The overall prevalence of gastrointestinal parasitic infections among captive wild mammals in mainland China is relatively high (53.9%). Significant differences in infection rates are observed across various seasons, regions, diagnostic methods, parasite species, and mammalian orders. Among the parasites, nematodes have the highest infection rate. Within the seven mammalian orders, primates exhibit the highest infection rate. No significant differences in infection rates are found among different dietary types or sampling times. The level of economic development in mainland China also exerts a significant influence on the parasitic infection rates of captive animals.

## Figures and Tables

**Figure 1 vetsci-12-00182-f001:**
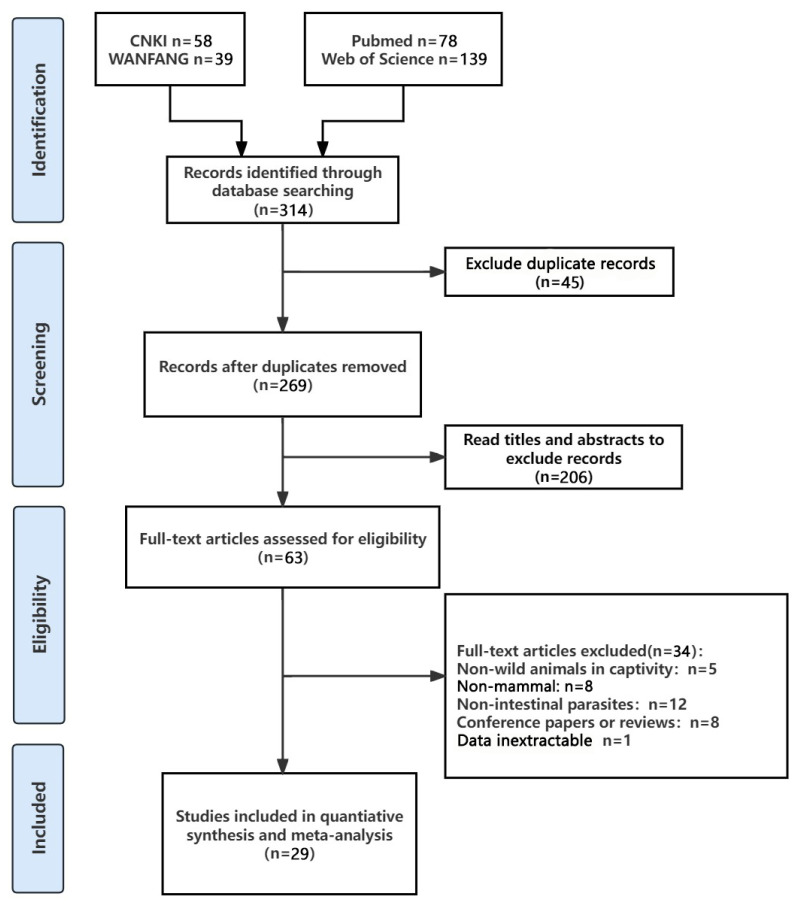
Flow diagram of the selection of eligible studies.

**Figure 2 vetsci-12-00182-f002:**
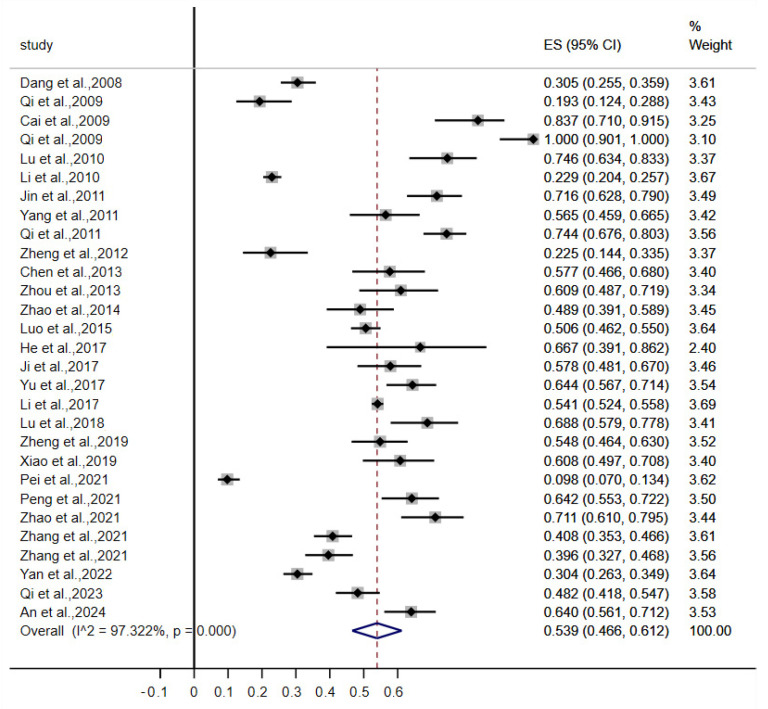
Forest plot depicting intestinal parasite infection among captive wild animals with random-effects analyses (ES, effect size; CI, confidence interval) [25,26,27,28,29,30,31,32,33,34,35,36,37,38,39,40,41,42,43,44,45,46,47,48,49,50,51,52,53].

**Figure 3 vetsci-12-00182-f003:**
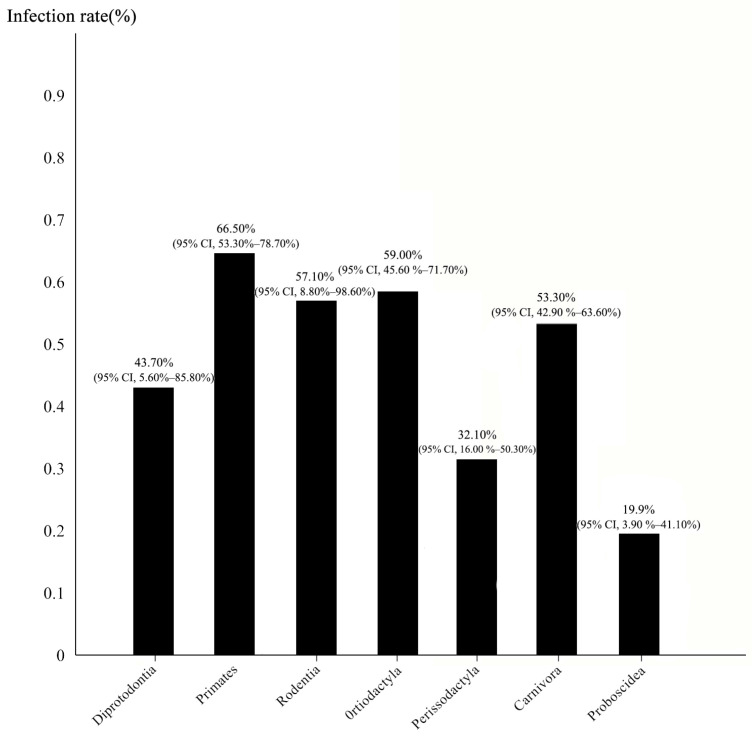
The infection status of intestinal parasites in captive wild mammals from different families.

**Figure 4 vetsci-12-00182-f004:**
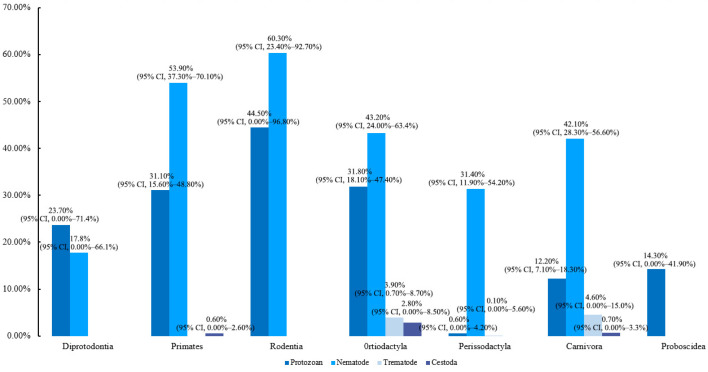
Intestinal parasite infection in captive wild mammals across different orders.

**Figure 5 vetsci-12-00182-f005:**
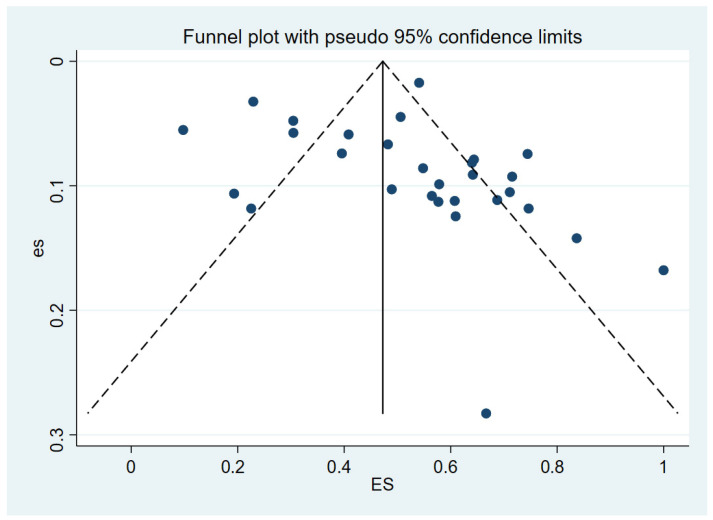
Funnel plot showing publication bias in studies reporting the intestinal parasite infection among captive wild mammals.

**Figure 6 vetsci-12-00182-f006:**
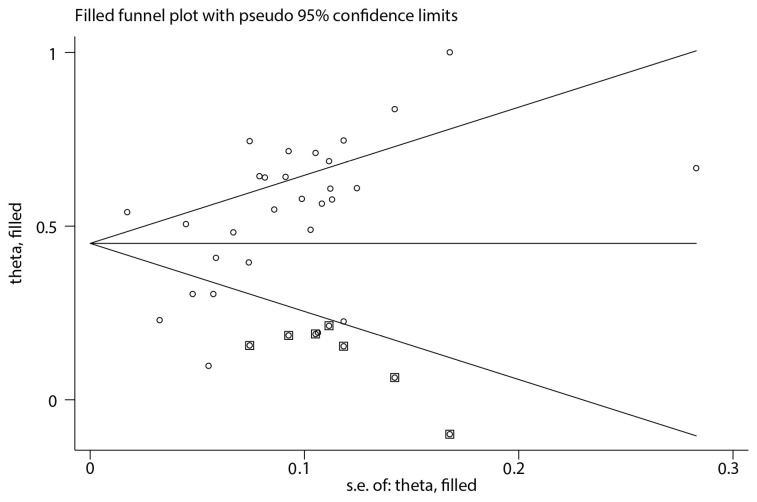
Funnel plot with trim and fill testing for publication bias. (dot: the actual studies; box: the imputed missing studies).

**Figure 7 vetsci-12-00182-f007:**
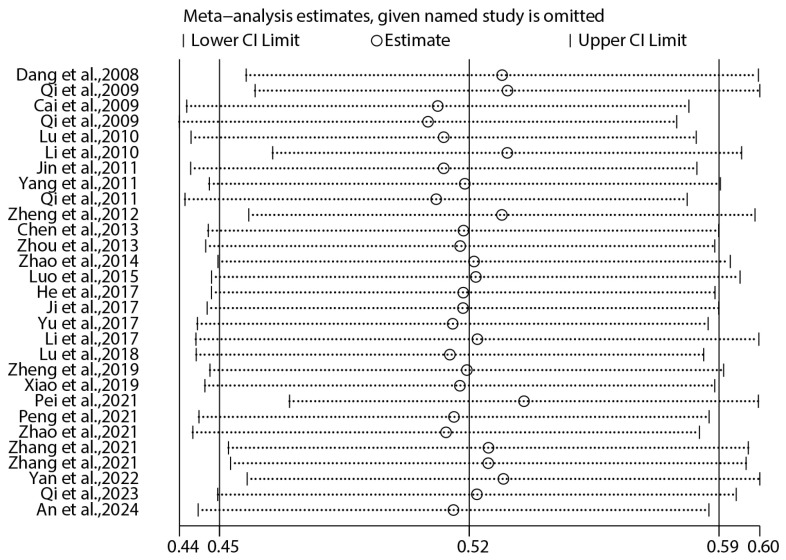
Influence of individual trials on pooled ES. The circles indicate the pooled relative risk estimate when each individual trial is omitted [25,26,27,28,29,30,31,32,33,34,35,36,37,38,39,40,41,42,43,44,45,46,47,48,49,50,51,52,53].

**Table 1 vetsci-12-00182-t001:** Characteristics of the eligible studies.

No	Authors and Publication Years	Study Years	Diagnostictechniques	Location	Sample Type	Sample Size	No. Positive	Parasites
1	Dang et al. [25]	2008	①, ④	Zoo	feces, stomach contents	302	92	Protozoan, Nematode
2	Qi et al. [26]	2009	①, ②	Zoo	feces	88	17	Protozoan, Nematode
3	Cai et al. [27]	2009	③, ⑤	Zoo	feces	49	41	Protozoan, Nematode, Nematode Cestoda
4	Qi et al. [28]	2009	①	Zoo	feces	35	35	Protozoan, Nematode
5	Lu et al. [29]	2010	③, ⑤	Zoo	feces	71	53	Protozoan, Nematode, Nematode
6	Li et al. [30]	2010	①, ②, ④	Zoo	feces	951	218	Protozoan, Nematode, Nematode
7	Jin et al. [31]	2011	①, ②	Zoo	feces	116	83	Protozoan, Nematode, Nematode
8	Yang et al. [32]	2011	①, ②, ⑤	Zoo, Animal Protection Center	feces	85	48	Protozoan, Nematode
9	Qi et al. [33]	2011	①, ②, ⑤	Zoo	feces	180	134	Protozoan, Nematode, Nematode, Cestoda
10	Zheng et al. [34]	2012	①, ②	Zoo	feces	71	16	Protozoan, Nematode, Cestoda
11	Chen [35]	2013	⑦, ⑨	Zoo	feces	78	45	Protozoan, Nematode, Nematode, Cestoda
12	Zhou et al. [36]	2013	①, ②, ⑤	Zoo	feces	64	39	Protozoan, Nematode
13	Zhao et al. [37]	2014	①, ②	Zoo	feces	94	46	Protozoan, Nematode
14	Luo et al. [38]	2015	①, ⑧	Zoo	feces	500	253	Protozoan, Nematode, Nematode, Cestoda
15	He et al. [39]	2017	①	Zoo	feces	12	8	Protozoan
16	Ji et al. [40]	2017	①, ③, ⑤	Zoo	feces	102	59	Protozoan, Nematode, Nematode
17	Yu [41]	2017	①, ②, ⑤	Zoo	feces	160	103	Protozoan, Nematode
18	Li et al. [42]	2017	①, ⑤	Zoo	feces	3349	1811	Protozoan, Nematode
19	Lu et al. [43]	2018	③, ⑤	Zoo	feces	80	55	Protozoan, Nematode
20	Zheng [44]	2019	①, ②	Zoo	feces	135	74	Protozoan, Nematode, Nematode
21	Xiao et al. [45]	2019	①, ②	Zoo	feces	79	48	Protozoan, Nematode, Nematode, Cestoda
22	Pei et al. [46]	2021	①, ②	Zoo	feces	328	32	Protozoan, Nematode
23	Peng et al. [47]	2021	①, ②, ⑨	Zoo	feces	120	77	Protozoan, Nematode
24	Zhao [48]	2021	①, ⑧	Siberian Tiger Park	feces	90	64	Protozoan, Nematode
25	Zhang et al. [49]	2021	⑥	Zoo	feces	289	118	Protozoan
26	Zhang et al. [50]	2021	⑥	Zoo	feces	182	72	Protozoan
27	Yan et al. [51]	2022	①, ②, ⑤	Zoo	feces	437	108	Protozoan, Nematode, Cestoda
28	Qi et al. [52]	2023	⑥	Zoo	feces	224	108	Protozoan
29	An et al. [53]	2024	①, ②	Zoo	feces	150	96	Protozoan, Nematode

Legend: ① Flotation method/saturated sugar solution flotation method/saturated salt solution flotation method; ② Sedimentation method/centrifugal sedimentation method/natural sedimentation method/water sedimentation method; ③ Direct smear method/physiological saline smear method; ④ Modified acid-fast staining method; ⑤ Iodine staining method/Lugol’s iodine staining method; ⑥ Nested PCR; ⑦ McMaster method; ⑧ Direct observation method; ⑨ Larval culture method.

**Table 2 vetsci-12-00182-t002:** Study quality assessment showing the number of the included studies in each category of a simple rating scale based on a checklist of six items.

	Items	No. of Included Studies for Each Category		
NO.		Yes Score 2	Unsure Score 1	No Score 0
1	Is the research question or objective clearly described and stated?	29	0	0
2	Are the sources of wild animals clearly specified?	28	0	1
3	Is the sampling time clearly stated?	25	0	4
4	Is the parasite detection method clearly and detailed described?	21	8	0
5	Is there detailed information on the detected intestinal parasites?	25	4	0
6	Is the classification of animals clearly specified?	15	12	2

**Table 3 vetsci-12-00182-t003:** Pooled estimates of the intestinal parasite infection of captive wild mammals by potential risk factors with meta-analysis.

Intestinal Parasite Infection	No. of Studies	Total No. of Animals	No. of Positive Animals	Infection Rate		Heterogeneity			*p*
				Estimates	95% CI	x^2^	PQ	I^2^ (%)	
Overall	29	8421	3978	53.90%	0.466–0.612	1045.578	0.0001	97.32%	
Group									
Season									
Spring	4	578	233	43.00%	0.244–0.626	59.563	0.0001	94.96%	0.038
Summer	2	81	56	61.80%	0.512–0.719	/	0.0001	/	
Autumn	3	589	230	50.00%	0.066–0.934	297.017	0.0001	99.33%	
Winter	3	229	138	61.60%	0.460–0.761	11.385	0.003	82.43%	
Year aroud	8	6010	2758	42.60%	0.320–0.535	377.219	0.0001	98.14%	
Region									
Northeast region	2	102	72	71.20%	0.616–0.799	/	0.0001	/	0.009
Eastern region	11	2876	1096	45.90%	0.338–0.583	422.143	0.0001	97.631	
Western region	8	2663	1458	56.90%	0.474–0.870	114.978	0.0001	93.91%	
Central region	13	2755	1304	51.50%	0.387–0.642	458.96	0.0001	97.39%	
Parasite									
Protozoan	27	7899	2239	27.40%	0.182–0.378	2044.973	0.0001	98.73%	0.0001
Nematode	25	7282	2365	45.10%	0.372–0.531	859.641	0.0001	97.21%	
Trematode	9	1185	65	5.50%	0.011–0.124	119.756	0.0001	93.32%	
Cestoda	9	1958	42	2.70%	0.013–0.046	23.193	0.003	65.51%	
Order Mammalia									
Diprotodontia	5	33	20	43.70%	0.056–0.858	7.737	0.052	61.23%	0.004
Primates	16	710	376	66.50%	0.533–0.787	142.484	0.0001	89.47%	
Rodentia	9	59	45	57.10%	0.088–0.986	60.218	0.0001	86.72%	
Artiodactyla	17	1322	636	59.00%	0.456–0.717	332.356	0.0001	95.19%	
Perissodactyla	10	137	47	32.10%	0.160–0.503	32.235	0.0001	72.08%	
Carnivora	17	774	413	53.30%	0.429–0.636	120.267	0.0001	86.70%	
Proboscidea	6	58	14	19.90%	0.039–0.411	5.086	0.279	21.35%	
Sampling years									
2007–2012	11	2085	793	50.90%	0.358–0.659	403.933	0.0001	97.52%	0.459
2013–2018	7	1052	584	57.70%	0.527–0.627	12.361	0.054	51.46%	
2019–2024	6	1349	510	46.70%	0.265–0.674	290.873	0.0001	98.28%	
Detection methods									
Microscopic techniques	26	7726	3680	55.30%	0.471–0.634	1036.759	0.0001	97.59%	0.012
PCR	3	695	298	42.90%	0.377–0.481	3.872	0.144	48.35%	
Feeding habits									
Carnivores	16	500	301	55.00%	0.436–0.661	75.003	0.0001	80.00%	0.971
Herbivores	18	1606	747	56.60%	0.451–0.678	322.585	0.0001	94.73%	
Omnivores	20	987	503	55.50%	0.444–0.664	204.329	0.0001	90.70%	

## Data Availability

Data are contained within the article.

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
