# Peer review of "The Occurrence and Meta-Analysis of Investigations on Intestinal Parasitic Infections Among Captive Wild Mammals in Mainland China"

_vetsci, 2025, doi:10.3390/vetsci12020182_

Round 1
Reviewer 1 Report
Comments and Suggestions for Authors
This study is a nice one. I have some suggestions and recommendations before publication.
Table 1. When you explain below the table the corresponding method for each number please write before „Legend”. Take care that for McMaster the number „7” is not in a circle as the others.
Figure 2. Put the legend in the brackets after the title of the figure.
Line 178. „, .... animal order, family, and are shown in (Table 3). ” – I think something must follow after „and”, and for sure not „are”.
Table 3. Please check the title, wildlife in captive? In case of year and method subgroups you have used in the first colomn the number of samples instead of number of studies.
Some discussions related to domestic animals are useless for this study. As an example see section from line 297 to line 301. What is the reletion of Cryptosporidiosis in Egyptian calves with the present study? Or the prevalence of coccidiosis in dogs from Pakistan?
Line 311. If you write the P value, you don’t have to write that it was lower than 0.05. I think it is obvious. Check and adjust all over the manuscript.
There is no discussion why the prevalence was higher in Suidae family followed by Bovidae
and in Canidae families. Then, the authors omitted to include among these families, Cercopithecid family with a calculated prevalence of 74.2%. Maybe it will be useful to specify also the less affceted families.
I recommend that the authors improve their paper by calculating the prevalence of intestinal parasites according to season for eack group of mammals, not really by family but maybe by using higher taxons as carnivores, omnivores, herbivores and so on.
Then, it will be interesting to present wich families of nematodes were more prevalent, and the same for protozoans and so on.
Author Response
Response to Reviewer 1 Comments
|
||||||||||||||||||||||||||||||||||||||||||||||||||||||||||||||||||||||||||||||||||||||||||||||||||||||||||||||||||||||||||||||||||||||||||||||||||||||||||||||||||||||||||||||||||||||||||||||||||||||||||||||||||||||||||||||||||||||||||||||||||||
1. Summary |
|
|
||||||||||||||||||||||||||||||||||||||||||||||||||||||||||||||||||||||||||||||||||||||||||||||||||||||||||||||||||||||||||||||||||||||||||||||||||||||||||||||||||||||||||||||||||||||||||||||||||||||||||||||||||||||||||||||||||||||||||||||||||
Thank you very much for taking the time to review this manuscript. Please find the detailed responses below and the corresponding revisions/corrections highlighted/in track changes in the re-submitted files.
|
||||||||||||||||||||||||||||||||||||||||||||||||||||||||||||||||||||||||||||||||||||||||||||||||||||||||||||||||||||||||||||||||||||||||||||||||||||||||||||||||||||||||||||||||||||||||||||||||||||||||||||||||||||||||||||||||||||||||||||||||||||
2. Questions for General Evaluation |
Reviewer’s Evaluation |
Response and Revisions |
||||||||||||||||||||||||||||||||||||||||||||||||||||||||||||||||||||||||||||||||||||||||||||||||||||||||||||||||||||||||||||||||||||||||||||||||||||||||||||||||||||||||||||||||||||||||||||||||||||||||||||||||||||||||||||||||||||||||||||||||||
Does the introduction provide sufficient background and include all relevant references? |
Yes |
|
||||||||||||||||||||||||||||||||||||||||||||||||||||||||||||||||||||||||||||||||||||||||||||||||||||||||||||||||||||||||||||||||||||||||||||||||||||||||||||||||||||||||||||||||||||||||||||||||||||||||||||||||||||||||||||||||||||||||||||||||||
Is the research design appropriate? |
Yes |
|
||||||||||||||||||||||||||||||||||||||||||||||||||||||||||||||||||||||||||||||||||||||||||||||||||||||||||||||||||||||||||||||||||||||||||||||||||||||||||||||||||||||||||||||||||||||||||||||||||||||||||||||||||||||||||||||||||||||||||||||||||
Are the methods adequately described? |
Yes |
|
||||||||||||||||||||||||||||||||||||||||||||||||||||||||||||||||||||||||||||||||||||||||||||||||||||||||||||||||||||||||||||||||||||||||||||||||||||||||||||||||||||||||||||||||||||||||||||||||||||||||||||||||||||||||||||||||||||||||||||||||||
Are the results clearly presented? |
|
|
||||||||||||||||||||||||||||||||||||||||||||||||||||||||||||||||||||||||||||||||||||||||||||||||||||||||||||||||||||||||||||||||||||||||||||||||||||||||||||||||||||||||||||||||||||||||||||||||||||||||||||||||||||||||||||||||||||||||||||||||||
Are the conclusions supported by the results? |
Yes |
|
||||||||||||||||||||||||||||||||||||||||||||||||||||||||||||||||||||||||||||||||||||||||||||||||||||||||||||||||||||||||||||||||||||||||||||||||||||||||||||||||||||||||||||||||||||||||||||||||||||||||||||||||||||||||||||||||||||||||||||||||||
3. Point-by-point response to Comments and Suggestions for Authors |
||||||||||||||||||||||||||||||||||||||||||||||||||||||||||||||||||||||||||||||||||||||||||||||||||||||||||||||||||||||||||||||||||||||||||||||||||||||||||||||||||||||||||||||||||||||||||||||||||||||||||||||||||||||||||||||||||||||||||||||||||||
Comments 1: [Table 1. When you explain below the table the corresponding method for each number please write before „Legend”. Take care that for McMaster the number „7” is not in a circle as the others.] Response 1: Thank you for pointing this out. We have corrected it to:⑦ McMaster method
|
||||||||||||||||||||||||||||||||||||||||||||||||||||||||||||||||||||||||||||||||||||||||||||||||||||||||||||||||||||||||||||||||||||||||||||||||||||||||||||||||||||||||||||||||||||||||||||||||||||||||||||||||||||||||||||||||||||||||||||||||||||
Comments 2: [Figure 2. Put the legend in the brackets after the title of the figure.] Response 2: We agree with your point and thank you for your kind comments. The legend has been placed in the brackets after the title of the figure: Figure 2. Forest plot depicting intestinal parasite infection among captive wild animals with random-effects analyses (ES, effect size; CI, confidence interval). |
||||||||||||||||||||||||||||||||||||||||||||||||||||||||||||||||||||||||||||||||||||||||||||||||||||||||||||||||||||||||||||||||||||||||||||||||||||||||||||||||||||||||||||||||||||||||||||||||||||||||||||||||||||||||||||||||||||||||||||||||||||
Comments 3: [Line 178. „, .... animal order, family, and are shown in (Table 3). ” – I think something must follow after „and”, and for sure not „are”.] |
||||||||||||||||||||||||||||||||||||||||||||||||||||||||||||||||||||||||||||||||||||||||||||||||||||||||||||||||||||||||||||||||||||||||||||||||||||||||||||||||||||||||||||||||||||||||||||||||||||||||||||||||||||||||||||||||||||||||||||||||||||
Response 3: We sincerely thank you for the useful suggestions. We have corrected it to “The subgroup analysis based on season, year, parasite category, feeding habits, order, and detection method are shown in (Table 3).” |
||||||||||||||||||||||||||||||||||||||||||||||||||||||||||||||||||||||||||||||||||||||||||||||||||||||||||||||||||||||||||||||||||||||||||||||||||||||||||||||||||||||||||||||||||||||||||||||||||||||||||||||||||||||||||||||||||||||||||||||||||||
Comments 4: [Table 3. Please check the title, wildlife in captive? In case of year and method subgroups you have used in the first column the number of samples instead of number of studies.] Response 4: We greatly appreciate your suggestion. Table 3 has been updated to “Pooled estimates of the intestinal parasite infection of captive wild mammals by potential risk factors with meta-analysis.” There was a line error when copying data from Excel to Word. We apologize for this and thank you for kind comments. We have corrected it to (Part of the table):
|
||||||||||||||||||||||||||||||||||||||||||||||||||||||||||||||||||||||||||||||||||||||||||||||||||||||||||||||||||||||||||||||||||||||||||||||||||||||||||||||||||||||||||||||||||||||||||||||||||||||||||||||||||||||||||||||||||||||||||||||||||||
Comments 5: [Some discussions related to domestic animals are useless for this study. As an example see section from line 297 to line 301. What is the reletion of Cryptosporidiosis in Egyptian calves with the present study? Or the prevalence of coccidiosis in dogs from Pakistan?] Response 5: Thank you for pointing this out. The animals were reclassified and analyzed based on their taxonomic orders, and the previous discussion section was removed. The revised discussion is as follows: In this study, significant differences in infection rates were found among animals from different taxonomic orders. Specifically, the highest infection rate of gastrointestinal parasites was found in the order Primates, at 66.5%. This rate is notably higher than the rates reported for non-human primates in zoological institutions in France [74] (53.9%) and for primates at Zoo Negara in Malaysia [75] (54.5%). As social animals, primates frequently interact within and between groups, facilitating parasite transmission. In mainland China, increased interactions between zoo visitors and primates may further contribute to the spread of gastrointestinal parasites. Additionally, the study found that the infection rate in Artiodactyla (59%) was higher than that in Perissodactyla (32.1%), consistent with findings from the Rio de Ja-neiro Zoo [54]. Variation in parasitic infection rates across different taxonomic orders may be influenced by differences in digestive physiology, social behaviors, environ-mental adaptability, and immune responses. For instance, Artiodactyla generally exhibit more complex social structures than Perissodactyla, which tend to have smaller group sizes and simpler social dynamics. Furthermore, Artiodactyla engage in rumination, and their multi-chambered stomachs harbor diverse microbial populations, potentially affecting parasite susceptibility [76]. Therefore, zoos should implement targeted parasite control measures tailored to the specific biological and behavioral characteristics of each animal order. Regarding different dietary habits, several studies have demonstrated that herbivorous animals tend to have a higher prevalence of gastrointestinal parasite infections than carnivorous animals [55, 77]. However, in our study, no significant difference in infection rates was observed among animals with different feeding habits, with only a marginal difference. This suggests that feeding habits of captive animals in mainland China do not significantly influence parasite infection rates. Comments 6: [Line 311. If you write the P value, you don’t have to write that it was lower than 0.05. I think it is obvious. Check and adjust all over the manuscript.] Response 6: We appreciate your suggestion. Items with P-values less than 0.05 have been removed from the manuscript. 1. There was a significant difference in parasitic infection rates among different seasons (p = 0.038). 2. There is significant heterogeneity in geography (p=0.009), with the highest infection rate in the Northeast region at 71.2% (95% CI, 61.6% -79.9%), 3. The infection rates of different species of parasites showed extremely significant differences (p=0.0001). 4. The analysis revealed significant differences in intestinal parasite infection rates among them. (p=0.004). 5.Subgroup analysis by sampling time publication year showed that the overall infection rate tended to decrease over time (Table 3), but the difference was not significant(p=0.459).
Comments 7 [There is no discussion why the prevalence was higher in Suidae family followed by Bovidae and in Canidae families. Then, the authors omitted to include among these families, Cercopithecid family with a calculated prevalence of 74.2%. Maybe it will be useful to specify also the less affceted families.] Response 7: Thank you for pointing this out. The previous discussion was reconsidered, and the classification of carnivores, omnivores, and herbivores was found to yield insignificant results. Consequently, these categories were merged for reanalysis. The original classifications were removed, and the discussion section was revised accordingly. In this study, significant differences in infection rates were found among animals from different taxonomic orders. Specifically, the highest infection rate of gastrointestinal parasites was found in the order Primates, at 66.5%. This rate is notably higher than the rates reported for non-human primates in zoological institutions in France [74] (53.9%) and for primates at Zoo Negara in Malaysia [75] (54.5%). As social animals, primates frequently interact within and between groups, facilitating parasite transmission. In mainland China, increased interactions between zoo visitors and primates may further contribute to the spread of gastrointestinal parasites. Additionally, the study found that the infection rate in Artiodactyla (59%) was higher than that in Perissodactyla (32.1%), consistent with findings from the Rio de Ja-neiro Zoo [54]. Variation in parasitic infection rates across different taxonomic orders may be influenced by differences in digestive physiology, social behaviors, environ-mental adaptability, and immune responses. For instance, Artiodactyla generally exhibit more complex social structures than Perissodactyla, which tend to have smaller group sizes and simpler social dynamics. Furthermore, Artiodactyla engage in rumination, and their multi-chambered stomachs harbor diverse microbial populations, potentially affecting parasite susceptibility [76]. Therefore, zoos should implement targeted parasite control measures tailored to the specific biological and behavioral characteristics of each animal order. Regarding different dietary habits, several studies have demonstrated that herbivorous animals tend to have a higher prevalence of gastrointestinal parasite infections than carnivorous animals [55, 77]. However, in our study, no significant difference in infection rates was observed among animals with different feeding habits, with only a marginal difference. This suggests that feeding habits of captive animals in mainland China do not significantly influence parasite infection rates.
Comments 8: [recommend that the authors improve their paper by calculating the prevalence of intestinal parasites according to season for each group of mammals, not really by family but maybe by using higher taxons as carnivores, omnivores, herbivores and so on.] Response 8: Thank you for your valuable suggestion. We followed three feeding habits classifications—carnivores, omnivores, and herbivores (feeding subgroups)—and subsequently reclassified the animals by order. The analysis section has also been updated in detail as mentioned in the previous comments. The updated data for this section is as follows:
Comments 9: [Then, it will be interesting to present which families of nematodes were more prevalent, and the same for protozoans and so on.] Response 9: Thank you for pointing this out. Based on your suggestions, we conducted a more detailed classification analysis of parasitic infections in different animal orders and supplemented our findings with corresponding figure 4 and results. The distribution of parasitic infections among different animal orders is illustrated in Figure 4. Overall, nematodes and protozoa exhibited the highest infection rates. Nematode infections were predominant in Primates, Rodentia, Artiodactyla, Perissodactyla, and Carnivora, whereas protozoan infections were more prevalent in Diprotodontia and Proboscidea. In contrast, trematodes and cestodes infection rates were comparatively low.
Figure 4. Intestinal parasite infection in captive wild mammals across different orders |
||||||||||||||||||||||||||||||||||||||||||||||||||||||||||||||||||||||||||||||||||||||||||||||||||||||||||||||||||||||||||||||||||||||||||||||||||||||||||||||||||||||||||||||||||||||||||||||||||||||||||||||||||||||||||||||||||||||||||||||||||||
|

Reviewer 2 Report
Comments and Suggestions for Authors
The study is very relevant; however, authors should improve methodological part, presentation of results and discussion, below are my comments.
Simple summary is too short, authors should describe the most important results/observations
L30 please add … and in winter 61.6%
L30-31 very similarly high values were determined in several other families like Cercopithecidae, Giraffidae, Bovidae and extremely low in Ailuridae.
L49 species [6, 7].
L64-67 Please correct the last sentence of the Introduction, now it is not clear, “the prevalence of,“?
Please check formatting of headings in Section 2 and Section 3
L76-78 could you please clarify why search words entered in Chinese and English slightly differed?
L89-90 What do you mean by “a third researcher intervened” why third, not second or forth
L97 not clear what is infection amount, please add. Is it prevalence, parasite load, abundance, richness or ?
L106-108 what about the validity of the method used? Also, in some published studies parasites are wrongly attributed to species or genus level. How did the authors address these issues? Or authors decided to restrict to higher systematic level Protozoa, Nematode, Trematoda, Cestoda?
Figure 1 Correct the text in some rectangles where there are no spaces between the words in the accents
Figure 3 and L195 in text “Suidae family at 85.8%”; however, in graphs prevalence do not exceed 80%.
L201 please add gap “significant(p=0.459”
Table 3 I do not agree with the statement that “overall infection rate tended to decrease over time”. In 2013-2018 period higher infection was observed than in 2007-2012, and the lowest rate in 2019-2024. Also, why overall infection rate is “0.539” rather than 53.90% as in separate years, seasons and families. It should be 42.60% rather thank 42.6. order of variables provided in different sampling years and using microscopic or PCR detection methods differs from what is provided above.
Authors should describe prevalence detected by PCR and microscopical methods. I find it surprising that lower infection rate was found by PCR compared to microscopy.
L210 “Part of the studies fell outside the confidence interval” please specify the number of such studies.
L237 please do not begin sentence with percentage
Discussion should be divided into subsections
In discussion there is a lack of explanation why infection rates are seemingly higher in summer compared to spring or autumn. Also, since study was performed in geographically broad area, so maybe temperatures and precipitation highly vary in the same season but in different regions?
L289 “2 literatures”?
L287-295 it should be expanded. Please describe in detail prevalence differences in regions examined and connect it with climate and developed economy showing higher prevention levels
L296 broader comparison should be made. Now, only the data obtained were compared only with three other studies
L310-311 move it to Results.
L310-319 please add citations. How authors could explain lower infection rates obtained using molecular methods vs traditional microscopical methods
L320-321 please explain why?
L331-334 Conclusions is written improperly, prevalence in Suidae is similar to that in several other families, very similar prevalence is obtained in summer and winter. Authors should rewrite this section, by highlighting the most important results and bringing them together in a focused way, now individual sentences without connective ideas are provided.
Comments on the Quality of English LanguageI have noticed several sentences that were not clear
L64-67
L89-90
L289
in principle, English does not interfere with the understanding of the text, but a minor revision of the language is required
Author Response
Response to Reviewer 2 Comments
|
||||||||||||||||||||||||||||||||||||||||||||||||||||||||||||||||||||||||
1. Summary |
|
|
||||||||||||||||||||||||||||||||||||||||||||||||||||||||||||||||||||||
Thank you very much for taking the time to review this manuscript. Please find the detailed responses below and the corresponding revisions/corrections highlighted/in track changes in the re-submitted files.
|
||||||||||||||||||||||||||||||||||||||||||||||||||||||||||||||||||||||||
2. Questions for General Evaluation |
Reviewer’s Evaluation |
Response and Revisions |
||||||||||||||||||||||||||||||||||||||||||||||||||||||||||||||||||||||
Does the introduction provide sufficient background and include all relevant references? |
Yes |
|
||||||||||||||||||||||||||||||||||||||||||||||||||||||||||||||||||||||
Is the research design appropriate? |
Yes |
|
||||||||||||||||||||||||||||||||||||||||||||||||||||||||||||||||||||||
Are the methods adequately described? |
Must be improved |
The Data Extraction section has been revised in accordance with the recommendations. |
||||||||||||||||||||||||||||||||||||||||||||||||||||||||||||||||||||||
Are the results clearly presented? |
Must be improved |
Figure 1, Figure 3, and Table 3 have been updated. Subgroup data for detection methods have been added. The taxonomic rank in the subgroup analysis has been changed from family to order. |
||||||||||||||||||||||||||||||||||||||||||||||||||||||||||||||||||||||
Are the conclusions supported by the results? |
Must be improved |
Based on the updated data, we have revised the Conclusions section. |
||||||||||||||||||||||||||||||||||||||||||||||||||||||||||||||||||||||
3. Point-by-point response to Comments and Suggestions for Authors |
||||||||||||||||||||||||||||||||||||||||||||||||||||||||||||||||||||||||
Comments 1: [L30 please add … and in winter 61.6% L30-31 very similarly high values were determined in several other families like Cercopithecidae, Giraffidae, Bovidae and extremely low in Ailuridae.] |
||||||||||||||||||||||||||||||||||||||||||||||||||||||||||||||||||||||||
Response 1: Thank you for pointing this out. We agree with this comment. The winter data has been added. According to the review comments, the classification was reclassified and analyzed based on order in the article. The updated section for comments 1 is as follows: Seasonal subgroup analysis revealed the highest incidence in summer at 61.8%, and 61.6% in winter. In the class order Mammalia, the highest infection rate was found in the Primates at 66.5% Suidae family at 85.8%and similarly high values were determined in several other orders like Artiodactyla (59%), Rodentia (57.1%), Carnivora (53.3%) and extremely low in Proboscidea (19.9%). |
||||||||||||||||||||||||||||||||||||||||||||||||||||||||||||||||||||||||
Comments 2: [L49 species [6, 7].] Response 2: Thank you for your suggestion. Excess symbols (.) have been removed.
|
||||||||||||||||||||||||||||||||||||||||||||||||||||||||||||||||||||||||
Comments 3: [L64-67 Please correct the last sentence of the Introduction, now it is not clear, “the prevalence of,“?] Response 3: Thank you for your kind comments. We have updated this sentence to: In this context, the present study therefore aimed to determine the prevalence and associated risk factors of intestinal parasites in captive wild mammals in Mainland China via a systematic review and meta-analysis. |
||||||||||||||||||||||||||||||||||||||||||||||||||||||||||||||||||||||||
Comments 4: [Please check formatting of headings in Section 2 and Section 3] Response 4: Thank you for pointing this out. We have modified the format.
|
||||||||||||||||||||||||||||||||||||||||||||||||||||||||||||||||||||||||
Comments 5: [L76-78 could you please clarify why search words entered in Chinese and English slightly differed?] Response 5: Thank you for pointing this out. We appreciate your comment on inaccurate translation of search term.,We have made modifications to ensure that the search terms used in the Chinese and English databases are consistent: The Chinese search terms were "parasites" or "intestinal parasites", "captive wild animals", "Zoo" or "Park". The English search terms were "parasites" or "intestinal parasites," "captive wild animal," "Zoo" or "Park" and "China."
|
||||||||||||||||||||||||||||||||||||||||||||||||||||||||||||||||||||||||
Comments 6: [L89-90 What do you mean by “a third researcher intervened” why third, not second or forth] Response 6: Thank you for your kind comments. The previous description may have been somewhat ambiguous. We have revised it to: If a disagreement or uncertainty arose about the eligibility of a study, the discrepancies were resolved by a third party or through discussion.
|
||||||||||||||||||||||||||||||||||||||||||||||||||||||||||||||||||||||||
Comments 7: [not clear what is infection amount, please add. Is it prevalence, parasite load, abundance, richness or] Response 7: Thank you for pointing this out. We have revised it to: ⑤ Sample size and positive size |
||||||||||||||||||||||||||||||||||||||||||||||||||||||||||||||||||||||||
Comments 8: [L106-108 what about the validity of the method used? Also, in some published studies parasites are wrongly attributed to species or genus level. How did the authors address these issues? Or authors decided to restrict to higher systematic level Protozoa, Nematode, Trematoda, Cestoda?] Response 8: We sincerely thank you for the useful suggestions. In the revised article, we have categorizing parasite according to higher taxonomic ranks, namely Protozoa, Nematoda, Trematoda, and Cestoda and performed comprehensive data statistics and analysis. In supplementary Data 2, the names and classifications of parasites are listed. |
||||||||||||||||||||||||||||||||||||||||||||||||||||||||||||||||||||||||
Comments 9: Figure 1 Correct the text in some rectangles where there are no spaces between the words in the accents] Response 9: We appreciate your reminder about error. Spaces have been added to the text: Exclude duplicate records.
|
||||||||||||||||||||||||||||||||||||||||||||||||||||||||||||||||||||||||
Comments 10: [Figure 3 and L195 in text “Suidae family at 85.8%”; however, in graphs prevalence do not exceed 80%.] Response 10: Thank you for your valuable suggestion. We have revised the taxonomic rank from family to order based on the reviewers’ comments and updated the corresponding content. |
||||||||||||||||||||||||||||||||||||||||||||||||||||||||||||||||||||||||
Comments 11: L201 please add gap “significant(p=0.459”] Response 11: We sincerely thank you for the suggestions. A space has been added. |
||||||||||||||||||||||||||||||||||||||||||||||||||||||||||||||||||||||||
Comments 12: I do not agree with the statement that “overall infection rate tended to decrease over time”. In 2013-2018 period higher infection was observed than in 2007-2012, and the lowest rate in 2019-2024. Also, why overall infection rate is “0.539” rather than 53.90% as in separate years, seasons and families. It should be 42.60% rather than 42.6. Order of variables provided in different sampling years and using microscopic or PCR detection methods differs from what is provided above. Authors should describe prevalence detected by PCR and microscopical methods. Authors should describe prevalence detected by PCR and microscopical methods. I find it surprising that lower infection rate was found by PCR compared to microscopy.] Response 12: Thank you for pointing this out. The data in Table 3 has been modified based on the classification of orders, and the results show that the subgroup results of sampling time are not significant. Therefore, the results of infection rate trends at different time periods have been deleted. The overall infection rate has been corrected to 53.90%, 42.6 also corrected to 42.60%. We appreciate your reminder about error. The data from PCR and traditional detection methods have been added to the results section
For the lower infection rate of PCR, we also put forward our viewpoint in the discussion section: There are significant differences between traditional detection methods and mole-cular detection in terms of detection results. Historically, the detection of intestinal parasites in literature mainly relied on microscopic examination of fecal samples. Microscopic parasite examination is highly dependent on the operator's experience and technical skills. Such reliance can lead to false positives due to subjective judgment [78, 79]. In the literature we reviewed, PCR has been used to detect parasites since its introduction in 2021. New diagnostic techniques offer more accurate, sensitive, and specific results. As a result, traditional detection methods often identify higher infection levels than PCR methods, which were introduced later. Given the advancements in molecular diagnostic techniques, it is recommended that molecular detection methods be prioritized for detecting gastrointestinal parasites in captive wild mammals, when conditions allow. |
||||||||||||||||||||||||||||||||||||||||||||||||||||||||||||||||||||||||
Comments 13: [ L210 “Part of the studies fell outside the confidence interval” Please specify the number of such studies.] Response 13: Thank you for your suggestion. Update as follows: A portion of the studies (15) fell outside the confidence interval |
||||||||||||||||||||||||||||||||||||||||||||||||||||||||||||||||||||||||
Comments 14: [L237 please do not begin sentence with percentage.] Response 14: Thank you for your suggestion. We have revised it to: Globally, 60% of terrestrial mammals are domesticated livestock, predominantly consisting of cattle and pigs. Humans account for 36% of the mammalian population, while wild mammals represent a mere 4%. |
||||||||||||||||||||||||||||||||||||||||||||||||||||||||||||||||||||||||
Comments 15: [In discussion there is a lack of explanation why infection rates are seemingly higher in summer compared to spring or autumn. Also, since study was performed in geographically broad area, so maybe temperatures and precipitation highly vary in the same season but in different regions?] Response 15: We greatly appreciate your suggestion. In response to the reviewers’ comments 15 and 18, we have segmented the discussion on seasonal variations and incorporated our perspectives on the higher infection rates observed in summer. Additionally, we have revised the discussion by integrating climate factors and regional differences as follows: Our research findings show significant variations in gastrointestinal parasite infection rates across different seasons and among regions within the same season. The highest infection rate was observed in summer at 61.8%, while winter also showed a relatively high infection rate of 61.6%. The high temperatures and humid conditions in summer create an optimal environment for the proliferation of mosquitoes, ticks, and other vectors, facilitating the expansion of intermediate hosts and transmission pathways, ultimately contributing to an increased parasite infection rate [68]. China's vast landmass exhibits significant geographical and climatic variations. Generally, the warm and humid climate of southern regions provides favorable conditions for parasite reproduction and transmission, leading to relatively high infection rates. In contrast, northern China is characterized by a temperate monsoon climate, while the northwest experiences an arid climate with scarce precipitation, both of which contribute to relatively lower parasite infection rates. However, the level of economic development in mainland China may exert a more significant influence on parasite infection [71]. Economically developed regions typically possess well-established infrastructure, including access to clean drinking water and improved sanitation. These regions also have better medical resources, such as specialized personnel, equipment, and medications, along with higher public awareness of health and hygiene [72]. Health education programs are more effectively implemented, and substantial investments are made in research related to parasite prevention and control [73]. In this study, the eastern regions of China, which are economically developed, exhibited the lowest parasite infection rate (45.9%). Conversely, the highest infection rate (71.2%) was observed in northeastern China, which may be attributed to lower levels of economic development. However, the relatively small sample size in this study necessitates further research to validate these findings. |
||||||||||||||||||||||||||||||||||||||||||||||||||||||||||||||||||||||||
Comments 16: [Figure 3 and L195 in text “Suidae family at 85.8%”; however, in graphs prevalence do not exceed 80%.] Response 16: Thank you for pointing out the errors in our mapping. We have categorized parasite according to higher taxonomic ranks and updated the corresponding content.
|
||||||||||||||||||||||||||||||||||||||||||||||||||||||||||||||||||||||||
Comments 17: [L289 “2 literatures”?] Response 17: We appreciate your reminder. We updated discussion content and revise it to:” (See Comments 15): …In this study, the eastern regions of China, which are economically developed, exhibited the lowest parasite infection rate (45.9%). Conversely, the highest infection rate (71.2%) was observed in northeastern China, which may be attributed to lower levels of economic development. However, the relatively small sample size in this study necessitates further research to validate these findings. |
||||||||||||||||||||||||||||||||||||||||||||||||||||||||||||||||||||||||
Comments 18: [L287-295 it should be expanded. Please describe in detail prevalence differences in regions examined and connect it with climate and developed economy showing higher prevention levels] Response 18: See comments 15.
|
||||||||||||||||||||||||||||||||||||||||||||||||||||||||||||||||||||||||
Comments 19: [L296 broader comparison should be made. Now, only the data obtained were compared only with three other studies] Response 19: We greatly appreciate your suggestion. A comparison was made based on broader (order).
|
||||||||||||||||||||||||||||||||||||||||||||||||||||||||||||||||||||||||
Comments 20: [L310-311 move it to Result] Response 20: We appreciate your suggestion to move it to the results section. Done.
|
||||||||||||||||||||||||||||||||||||||||||||||||||||||||||||||||||||||||
Comments 21: [L310-319] please add citations. How authors could explain lower infection rates obtained using molecular methods vs traditional microscopical methods Response 21: Thank you for your valuable suggestion. We have added literature and are discussing and presenting our views on the lower infection rate of PCR. See comments 12: …Such reliance can lead to false positives due to subjective judgment [78, 79].
|
||||||||||||||||||||||||||||||||||||||||||||||||||||||||||||||||||||||||
Comments 22: [L320-321 please explain why?] Response 22: Thank you for your suggestion. Mainland China has a vast territory. This study covers most but not all regions of China. We will revise it to:” The current methodology may not comprehensively capture all gastrointestinal parasitic infections in captive wild mammals in China. ”
|
||||||||||||||||||||||||||||||||||||||||||||||||||||||||||||||||||||||||
Comments 23: [L331-334 Conclusions is written improperly, prevalence in Suidae is similar to that in several other families, very similar prevalence is obtained in summer and winter. Authors should rewrite this section, by highlighting the most important results and bringing them together in a focused way, now individual sentences without connective ideas are provided. ] Response 23: Thank you for your kind suggestion. We have conclusion refined and rewritten it: The overall prevalence of gastrointestinal parasitic infections among captive wild mammals in mainland China is relatively high (53.9%). Significant differences in infection rates are observed across various seasons, regions, diagnostic methods, parasite species, and mammalian orders. Among the parasites, nematodes have the highest infection rate. Within the seven mammalian orders, primates exhibit the highest infection rate. No significant differences in infection rates are found among different dietary types or sampling times. The level of economic development in mainland China also exerts a significant influence on the parasitic infection rates of captive animals. |
||||||||||||||||||||||||||||||||||||||||||||||||||||||||||||||||||||||||
4. Response to Comments on the Quality of English Language |
||||||||||||||||||||||||||||||||||||||||||||||||||||||||||||||||||||||||
Point 1: L64-67 |
||||||||||||||||||||||||||||||||||||||||||||||||||||||||||||||||||||||||
Response 1: We will revise it to: In this context, the present study therefore aimed to determine the prevalence and associated risk factors of intestinal parasites in captive wild mammals in mainland China via a systematic review and meta-analysis. |
Point 1: L89-90 |
Response 1: We will revise it to: If a disagreement or uncertainty arose about the eligibility of a study, the discrepancies were resolved by a third party or through discussion. |
Point 1: L289 Although the infection rate is highest in the Northeast region, 2 literatures was only in summer and autumn, which has certain limitations in the results. |
Response 1: We have updated this section and revise it to:” (See Comments 15): …In this study, the eastern regions of China, which are economically developed, exhibited the lowest parasite infection rate (45.9%). Conversely, the highest infection rate (71.2%) was observed in northeastern China, which may be attributed to lower levels of economic development. However, the relatively small sample size in this study necessitates further research to validate these findings. |

Round 2
Reviewer 1 Report
Comments and Suggestions for Authors
The manuscript was considerably improved.
Two minor observations.
Table 1. The authors did not introduce/answer to the suggestion; I mean under the table they have to write „Legend” before explaining numbers.
Line 304. Do not put Table 3 in brackets. „....and detection method is shown in Table 3.” Instead of „......and detection method is shown in (Table 3).”
Author Response
Comments 1: [Two minor observations. Table 1. The authors did not introduce/answer to the suggestion; I mean under the table they have to write „Legend” before explaining numbers.] Response 1: Thank you for your suggestion. „Legend "has been added before the number, as follows: Legend: â‘ Flotation method/saturated sugar solution flotation method/saturated salt solution flotation method; â‘¡……
|
Comments 2: [Line 304. Do not put Table 3 in brackets. „....and detection method is shown in Table 3.” Instead of „......and detection method is shown in (Table 3).”] Response 2: Thank you for your kind comments. Brackets have been deleted:The subgroup analysis based on season, year, parasite category, feeding habits, order, and detection method is shown in Table 3.
|

Reviewer 2 Report
Comments and Suggestions for Authors
I still believe that simple summary is too short, authors should describe the most important results/observations. How many articles (approximately) were chosen to analyse, that corresponds to criteria set. Please add in which families of animal’s highest infection rates was determined (not including actual figures). Also, authors can add which parasite group was most common or in which season highest prevalence was observed.
Abstract is well modified
L67 maybe “better about one of tenth” instead of 11%, since this figure is not round neither 10 or 15.
L88-90 the sentence is not correct in English. Please use due to or as a result of
L185-186 please check formatting “parasites(protozoa/nematodes/flukes/tapeworms)”
L186 the dot after the end of the sentence is missing
L199 please check formatting “â‘¥Is the classification”
Figure 3 and Figure 4. Please add confidence intervals on bars. Since no additional fee is applied for coloured figures, my suggestion is to make the fourth picture in colour.
L737-739 Whether this idea is new, suggested by you, or whether you should cite other articles?
Discussion and conclusions are seemingly improved
Author Response
Comments 1: [I still believe that simple summary is too short, authors should describe the most important results/observations. How many articles (approximately) were chosen to analyse, that corresponds to criteria set. Please add in which families of animal’s highest infection rates was determined (not including actual figures). Also, authors can add which parasite group was most common or in which season highest prevalence was observed.] Response 1: We sincerely thank you for the useful suggestions and improve the summary. To ensure the welfare of captive wild animals and public health, it is essential to assess and monitor the gastrointestinal parasitic infection status of captive wild mammals. This study presents the first meta-analysis and systematic review to evaluate the prev-alence of gastrointestinal parasitic infections among captive wild mammals in main-land China and the associated risk factors. A total of 29 eligible studies were included in this comprehensive analysis. The findings revealed a high prevalence of gastrointestinal parasitic infections among captive wild mammals in mainland China, with nematodes identified as the most dominant parasite group. Among taxonomic orders, primates exhibited the highest infection rates, followed by Artiodactyla and Rodentia, whereas Proboscidea showed the lowest prevalence. Seasonally, infection rates peaked during summer and winter, likely due to favorable environmental conditions and behavioral patterns of captive animals. Furthermore, infection prevalence was lower in economically developed regions, suggesting that improved sanitation and management practices play a critical role in mitigating parasite transmission. We emphasize the need for a sanitation management program and regular antiparasitic treatment in captive wild animals.
|
Comments 2: [L67 maybe “better about one of tenth” instead of 11%, since this figure is not round neither 10 or 15.] Response 2: Thank you for your kind comments. We revised it to: China is home to approximately one of tenth of the world's total wild mammal species [1].
|
Comments 3: [L88-90 the sentence is not correct in English. Please use due to or as a result of] |
Response 3: We greatly appreciate your suggestion. We revised it to: Zoonotic parasites are likely to be transmitted from animals to human due to direct contact between animal caretakers or tourists and wild animals.
Comments 4: [L185-186 please check formatting“parasites(protozoa/nematodes/flukes/tapeworms)”] Response 4: Thank you for your suggestion. The format and updates are as follows: â‘£ Name and classification of detected parasites (protozoan or nematode or trematode or cestoda). In addition, we revised “…question/objective…” to“… or…” in Table 2. No 1. Comments 5: [L186 the dot after the end of the sentence is missing] Response 5: Sorry for my mistake and thank you for pointing this out. The dot has been added and other punctuation marks in this part have been updated. The dot has been added and other 2 punctuation marks in this section have been updated. â‘£ Name and classification of detected parasites (protozoan or nematode or trematode or cestoda). ⑤ Sample size and positive size.
Comments 6: [L199 please check formatting “â‘¥Is the classification”] Response 6: Thank you for pointing this out. We modified the format and added spaces after â‘¥: â‘¥ Is the classification
Comments 7 [Figure 3 and Figure 4. Please add confidence intervals on bars. Since no additional fee is applied for coloured figures, my suggestion is to make the fourth picture in colour.] Response 7: Thank you for your kind comments. We added confidence intervals to Figure 3 and Figure 4, and made Figure 4 into a color chart. Comments 8: [L737-739 Whether this idea is new, suggested by you, or whether you should cite other articles?] Response 8: We appreciate your comments on this. This idea was put forward by us, and we revised the content appropriately: Considering the advancements in molecular diagnostic technology, we recommend prioritizing molecular detection methods or experienced clinical personnel for the diagnosis of gastrointestinal parasites in captive wild mammals, provided that the necessary resources and conditions are available. |
|
